# Mind the gap: Mapping variation between national and local clinical practice guidelines for acute paediatric asthma from the United Kingdom and the Netherlands

**Charlotte Koldeweij**[1,2]*, **Nicholas Appelbaum**[2,3], **Carmen Rodriguez Gonzalvez**[2], **Joppe Nijman**[4], **Ruud Nijman**[5], **Ruchi Sinha**[6], **Ian Maconochie**[7], **Jonathan Clarke**[3,8]

1 Radboud Institute for Health Sciences, Radboud University Medical Center, Nijmegen, The Netherlands, 2 Helix Centre for Design in Healthcare, Imperial College London, London, United Kingdom, 3 Department of Surgery and Cancer, Imperial College London, London, United Kingdom, 4 Department of Pediatric Intensive Care, University Medical Center Utrecht, Utrecht, The Netherlands, 5 Faculty of Medicine, Department of Infectious Diseases, Section of Paediatric Infectious Diseases, Imperial College London, London, United Kingdom, 6 Department of Paediatric Intensive Care, Division of Women and Children's Services, Imperial College Healthcare NHS Trust, London, United Kingdom, 7 Centre for Paediatrics and Child Health, Imperial College Healthcare NHS Trust, London, United Kingdom, 8 Centre for Mathematics of Precision Healthcare, Department of Mathematics, Imperial College London, London, United Kingdom

* charlotte.koldeweij@radboudumc.nl

## Abstract

### Background

Clinical practice guidelines (CPGs) aim to standardize clinical care. Increasingly, hospitals rely on locally produced guidelines alongside national guidance. This study examines variation between national and local CPGs, using the example of acute paediatric asthma guidance from the United Kingdom and the Netherlands.

### Methods

Fifteen British and Dutch local CPGs were collected with the matching national guidance for the management of acute asthma in children under 18 years old. The drug sequences, routes and methods of administration recommended for patients with severe asthma and the tone of recommendation across both types of CPGs were schematically represented. Deviations from national guidance were measured. Variation in recommended doses of intravenous salbutamol was examined. CPG quality was assessed using the Appraisal of Guidelines for Research and Evaluation (AGREE) II.

### Results

British and Dutch national CPGs differed in the recommended drug choices, sequences, routes and methods of administration for severe asthma. Dutch national guidance was more rigidly defined. Local British CPGs diverged from national guidance for 23% of their recommended interventions compared to 8% for Dutch local CPGs. Five British local guidelines and two Dutch local guidelines differed from national guidance for multiple treatment steps. Variation in second-line recommendations was greater than for first-line recommendations

**Data Availability Statement:** All relevant data are within the manuscript and its Supporting Information files.

**Funding:** Two authors (NA and CK) had support from the National Institute for Health Research (NIHR, accessible from https://www.nihr.ac.uk/) Imperial Patient Safety and Translational Research Centre (PSTRC) PSTRC_2016_004. Infrastructure support for this work was provided by the NIHR Imperial Biomedical Research Centre (BRC) 1215-20013. JC acknowledges support from the Engineering and Physical Sciences Research Council (EPSRC, accessible from: https://epsrc.ukri.org/) grant EP/N014529/1 supporting the EPSRC Centre for Mathematics of Precision Healthcare and from the Wellcome Trust (accessible from: https://wellcome.org/), grant 215938/Z/19/Z. RN is the recipient of the NIHR Academic clinical fellowship and lectureship award NIHR ACL 2018-021-007. The funders had no role in study design, data collection and analysis, decision to publish, or preparation of the manuscript.

**Competing interests:** All authors have completed the ICMJE uniform disclosure form at www.icmje.org/coi_disclosure.pdf and declare: two authors (NA and CK) had support from the National Institute for Health Research and one author (JC) had support from the Wellcome Trust for the submitted work. NA is the CEO of Dosium, a software company building decision support tools for medication safety. Authors had no financial relationships with any organisations that might have an interest in the submitted work in the previous three years; no other relationships or activities that could appear to have influenced the submitted work.

across local CPGs from both countries. Recommended starting doses for salbutamol infusions varied by more than tenfold. The quality of the sampled local CPGs was low across all AGREE II domains.

## Conclusions

Local CPGs for the management of severe acute paediatric asthma featured substantial variation and frequently diverged from national guidance. Although limited to one condition, this study suggests that unmeasured variation across local CPGs may contribute to variation of care more broadly, with possible effects on healthcare quality.

## Introduction

The increasing volume of new clinical evidence hinders the ability of individual clinicians to incorporate large amounts of constantly updated information into their decision-making [1]. Clinical practice guidelines (CPGs) are documents that systematically distil medical evidence into treatment recommendations to inform patient care [2, 3]. CPGs support the provision of evidence-based medicine and aim to optimize the quality and cost-effectiveness of care [1, 4–6]. CPGs are produced by numerous leading clinical institutions, including professional bodies and national societies like the National Institute for Health and Care Excellence (NICE) in the United Kingdom (UK) and the Dutch Federation of Medical Specialists [7–9]. These organisations systematically review the scientific literature, condensing the best available evidence into practice-oriented recommendations for various conditions [4, 10–13]. Expert opinion and consensus judgment may be used in the absence of conclusive evidence [10].

Despite their stated goal of standardising the quality of clinical care [3], CPGs are not a matter 'of one size fits all' [14]. It may not always be possible for clinicians at a given hospital to adopt national recommendations without adaptation to their local resources and needs, for example, differences in the availability of medications, various levels of local expertise or financial constraints [12, 14]. In contrast to the well-defined methodologies used to design national CPGs, the production of local CPGs remains a largely ad-hoc process, driven by small groups of clinicians with various levels of familiarity with guideline development [10, 15]. In addition to drawing on national CPGs, local CPGs may incorporate alternative sources of information, including locally available expert opinion and scientific literature published more recently than that incorporated in national CPGs [1, 16].

Although national CPGs and their locally adapted counterparts have become a staple of day-to-day care, variation in clinical practice continues to undermine healthcare outcomes [17, 18]. An example of this is provided by the management of acute paediatric asthma (APA). Asthma is the most common chronic paediatric condition in the UK and the Netherlands [19, 20]. There are approximately 25,000 annual emergency hospital admissions for APA in the UK [20]. Despite APA being extensively covered in national and local clinical guidance [21–23], variation in the emergency clinical management of APA is widespread [20, 24–26]. Significant variation in regional mortality from asthma, including APA, has been reported in recent years [27]. Variation in clinical practice has been related to differences in CPGs, including those produced by individual hospitals [28–30]. The nature and extent of local CPG variation, however, remains poorly understood. This study examines variation in the treatment recommendations for severe APA outlined in national guidance from the UK and the Netherlands and by a sample of local CPGs from British and Dutch hospitals.

## Methods

### Guideline and data sampling

National guidelines on APA produced by the British Thoracic Society/Scottish Intercollegiate Guideline Network (BTS/SIGN) in 2019 and the Dutch Paediatric Society (NVK) in 2012 were collected [21, 23]. Both documents were chosen as the clinical guidelines from the relevant learned bodies of the UK and the Netherlands for the management of acute paediatric asthma. A convenience sample of local CPGs for APA management were obtained between 1 January and 1 February 2019 from Dutch and British tertiary-level hospitals. Local CPGs were retrieved online from hospital websites where available, or were obtained from clinicians working at each hospital. The most recent version of each local CPG was used. CPG recommendations were subdivided according to the age of the patient, to the clinical severity of APA at presentation and according to a patient's response to treatment. Treatments recommended in children two to eighteen years old were examined. The criteria outlined by each CPG to assess asthma severity at presentaton were collected. Pharmacological interventions recommended for managing patients with the most severe form of asthma at presentation and by a lack of response to consecutive treatments were examined. Hereafter, this is referred to as the '*most severe*' treatment pathway. For each CPG, the dose, route (e.g. intravenous) and method (e.g. a bolus) of administration of each recommended drug was extracted, alongside the order in which it was suggested to be administered. The tone of each recommendation was noted and categorized as either mandatory ('*give*') or optional ('*consider giving*').

### Variation in drug choice, sequence and tone

Differences in drug choice, sequence and route of administration across the sampled local CPG treatment pathways were measured using the relevant national CPG as a reference. Local CPGs were awarded one point at each step if they recommended a drug type or an administration route that differed from their respective national CPG at that position or if additional alternatives were offered. Differences in methods of administration or in the tone of recommendation were not scored. Differences in drug choice, sequence, route and method of administration and tone of recommendation across the sampled treatment pathways were also represented in a common schematic diagram. Design idioms were borrowed from familiar urban transport maps. Each guideline was represented as a single line featuring the sequence of drugs recommended for a patient with most severe APA. Where a guideline offered clinicians the choice between two drugs with no distinct preference, both choices were represented. Drugs recommended for treating comorbidities, differential diagnoses or side effects resulting from the primary APA treatment were not represented.

### Variation in drug dosage

Beyond differences in recommended drug choice, sequence and tone of recommendation, variation may also exist in medication dosing. For APA, this has, for instance, been shown to be the case for continuous salbutamol infusions [31]. As a case study, the upper and lower limits of recommended salbutamol infusion dose ranges were compared across the sample.

### Guideline quality assessment

The quality of the sampled national and local guidelines from the UK and the Netherlands was scored independently by two reviewers (CK and RN) using the Appraisal of Guidelines Research & Evaluation (AGREE) II instrument [32]. AGREE II is a standardized tool that allows to assess the methodological quality of CPGs based on six domains: 1) scope and

purpose, 2) stakeholder involvement, 3) rigour of development, 4) clarity of presentation, 5) applicability and 6) editorial independence of the guideline. Each domain is comprised of several items that can each be scored on a 7-point scale taking into account predefined criteria. Discrepancies of over 3 points per item across reviewers were discussed and scores were revised as felt to be reasonable. Scaled domain scores were defined in line with AGREE II, by summing up item scores within each domain for each reviewer and standardizing it as a percentage of the maximum possible score [32].

## Results

### Guideline description

Of the sampled local CPGs, seven came from British tertiary-level hospitals and seven from Dutch tertiary-level hospitals, including two CPGs from different units of the same Dutch hospital (NL1a and NL1b). One additional Dutch hospital had no local CPG and used national (NVK) guidance (NL5). The basic characteristics of the sampled CPGs and the criteria used by each CPG to characterise APA severity at presentation are shown in Table 1. All CPGs stratified patients into two ('moderate' and 'severe') or three ('moderate', 'severe', and 'life-threatening') categories. In line with NVK guidance, most local Dutch CPGs distinguished two groups at presentation using a 94% oxygen saturation (SpO2) threshold. Local British CPGs mostly aligned with British national (BTS/SIGN) guidance, and defined three groups based on a 92% SpO2 threshold alongside other diagnostic parameters including the presence of signs of life-threatening APA.

### Variation in drug choice, sequence and tone

The recommended drug treatment pathways for each CPG are represented schematically in Fig 1. Differences in drug choice, sequence and route of administration across the sampled CPGs were quantified in Table 2.

The differences across nationally recommended pathways were examined. The NVK recommended six drugs in total, all of which were also recommended by BTS/SIGN. Alongside the drugs recommended by the NVK, BTS/SIGN also recommended nebulized magnesium sulfate and intravenous aminophylline. In both CPGs, the sequence and tone of recommendations differed more for second-line drugs than for first-line treatments. BTS/SIGN and the NVK recommended the same mandatory sequence of first-line treatments (oxygen, nebulized salbutamol, nebulized ipratropium bromide, oral corticosteroids) up to nebulized magnesium sulfate. The latter drug was advised as an optional treatment by British national guidance. National recommendations on second-line drugs diverged in both sequence and tone. While the NVK recommended *considering* intravenous magnesium sulfate before *giving* intravenous salbutamol, BTS/SIGN recommended that clinicians *consider* administering any of three recommended second-line drugs (intravenous magnesium sulfate, intravenous salbutamol and intravenous aminophylline) in the order they deemed most appropriate based on their assessment of the risks and benefits involved.

Local CPG recommendations for most severe APA were compared to the relevant national CPGs. All Dutch and British local CPGs aligned with their respective national CPG on four drugs. Three drugs were first-line treatments (oxygen, nebulized salbutamol, corticosteroids) and one was a second-line intervention (intravenous salbutamol). One British CPG (UK4) recommended nebulized terbutaline as an alternative for nebulized salbutamol without justifying this deviation from national guidance (see S1 File on the local considerations behind various sequences of second-line treatments). Six out of seven local British CPGs departed from BTS/SIGN recommendations by omitting nebulized magnesium sulfate. None of them provided a

**Table 1. Description of sampled acute paediatric asthma CPGs from the UK and the Netherlands.**

| Guideline | Last updated | Scale of application | Setting | Online access | Authors | Description of methods | Number of stratified patient groups at presentation | Assessment parameters for most severe APA at presentation |
|---|---|---|---|---|---|---|---|---|
| BTS/ SIGN | 2019 | National | Hospital including PICU | Yes | Paediatricians, respiratory consultants, pharmacists, researchers, nurse specialists & practictioners | Yes | Three | spO2 < 92% plus any of the following: PEF < 33% of best or predicted,[1] silent chest, cyanosis, poor respiratory effort, agitation, confusion |
| NVK | 2012 | National | Hospital, not specified | Yes | Paediatricians, paediatric respiratory consultants | Yes | Two[1] | spO2 ≤ 94% |
| UK1 | 2014 | Local | Hospital excluding PICU | No | Asthma Steering Group, not further specified | No | Three | spO2 < 92% plus any of the following: PEF < 33% of best or predicted value[1] silent chest and/ or cyanosis, poor respiratory effort, altered consciouness |
| UK2 | 2016 | Local | Not specified | No | Paediatrician | No | Three | spO2 < 92% plus any of the following: PEF < 33% of best or predicted value,[1] silent chest, cyanosis, poor respiratory effort, agitation or exhaustion, altered consciousness |
| UK3 | 2018 | Local | Ward excluding PICU | Yes | Paediatric respiratory consultant, paediatrician, clinical pharmacist | Yes | Three | Any of the following: spO2 < 92%, silent chest, cyanosis, poor respiratory effort, fatigue or exhaustion, agitation or reduced consciousness |
| UK4 | 2015 | Local | Not specified | No | Paediatric respiratory consultant, paediatrician, paediatric intensive care consultant, paediatric medical pharmacist | Yes | Three | spO2 < 92% plus any of the following: silent chest, cyanosis, poor respiratory effort, agitation or altered consciousness, exhaustion, increased pCO2 or hypotension[3] |
| UK5 | 2017 | Local | Hospital including PICU | No | Paediatric respiratory consultant, paediatrician, paediatric medical pharmacist | Yes | Three | Any of the following: spO2 < 92%, PEF < 33% of best or predicted value,[1] silent chest, poor respiratory effort, agitation or altered consciousness, heart rate ≥ 140/min (2–5 years) or > 125/min (≥ 5 years) |
| UK6 | 2017 | Local | Hospital, excluding PICU | Yes | Paediatric respiratory consultant, paediatricians, senior staff nurse | No | Two[1] | Any of the following: spO2 < 92%, PEF < 33% of predicted value,[1] silent chest, cyanosis, exhaustion with poor respiratory effort, confusion, maximal accessory muscle use/recession, unable to talk, coma, hypotension, marked tachycardia |
| UK7 | 2018 | Local | Not specified | No | Paediatric respiratory consultant, paediatrician, paediatric medical pharmacist | No | Three | spO2 < 92% plus any of the following: PEF < 33% of best or predicted,[1] silent chest, cyanosis, poor respiratory effort, fatigue or exhaustion, altered consciousness |
| NL1a | 2019 | Local | PICU | No | Unspecified | No | Two[1] | Not specified |
| NL1b | 2016 | Local | Hospital, incl. A&E | No | Paediatric respiratory consultants, paediatrician, | No | Two[1] | spO2 ≤ 94% |

(*Continued*)

**Table 1.** (Continued)

| Guideline | Last updated | Scale of application | Setting | Online access | Authors | Description of methods | Number of stratified patient groups at presentation | Assessment parameters for most severe APA at presentation |
|---|---|---|---|---|---|---|---|---|
| NL2 | 2017 | Local | Hospital incl. PICU | No | Paediatrician, Paediatric intensive care consultant, paediatric respiratory consultant | No | Two[1] | spO2 ≤ 94% |
| NL3 | Unknown | Local | Hospital incl. PICU | No | Paediatric respiratory consultant | No | Two[1] | spO2 ≤ 94% |
| NL4 | 2017 | Local | PICU | No | Unspecified | No | Two[1] | spO2 ≤ 94% |
| NL5 | Uses national CPG | | | | | | | |
| NL6 | Unknown | Local | Hospital | No | Unspecified | No | Two[1] | spO2 ≤ 94% |
| NL7 | 2011 | Local | Hospital | No | Paediatrician, paediatric intensive care consultant | No | Two | spO2 ≤ 94% |

A&E: Accidents & Emergency; APA: acute paediatric asthma, BTS/SIGN: British Society/Scottish Intercollegiate Guidelines Network, NL: Netherlands, NVK: Nationale Vereniging voor Kindergeneeskunde (Dutch Society for Paediatrics), PEF: peak expiratory flow, PICU: Paediatric Intensive Care Unit, SpO2: oxygen saturation, UK: United Kingdom.

[1] Applicable for children above 5 years.

rationale for this omission. Six Dutch local CPGs recommended the same drugs as the NVK. Differing with national recommendations, one Dutch local CPG omitted ipratropium bromide (NL1a), while two Dutch local CPGs included intravenous theophylline as their last recommended intervention (NL1a and NL2). These departures from national guidance were justified within the relevant local CPG.

Looking at the advised routes and methods of administration for recommended drugs, two Dutch and six British local CPGs deviated from national CPGs by recommending intravenous corticosteroids instead of, or following the administration of oral corticosteroids. A further

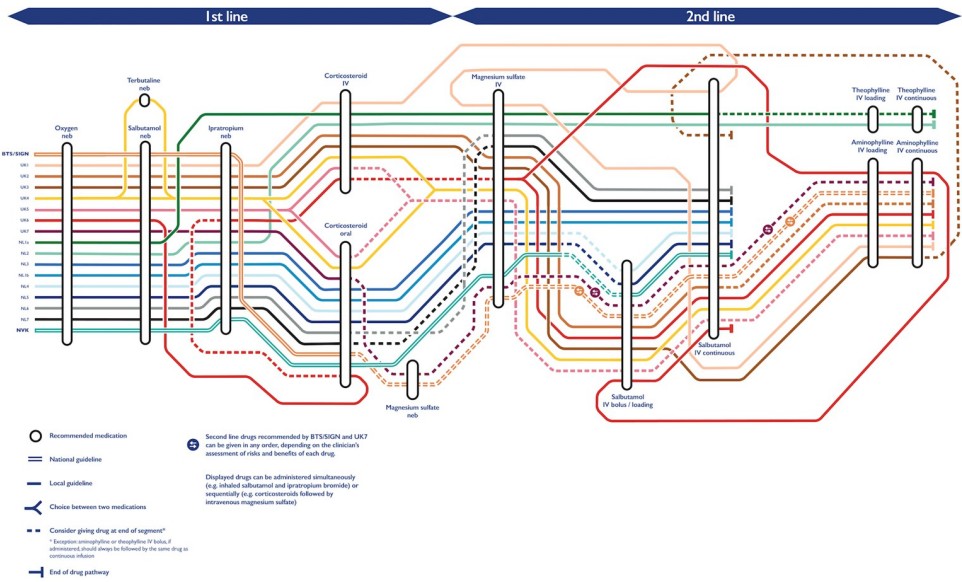

**Fig 1. Variation across treatment pathways for most severe acute paediatric asthma across British and Dutch CPGs.** Drug recommendations proceed from left to right. Iv: intravenous, neb: nebulized.

**Table 2. Quantitative assessment of deviations from national guidance across sampled local British and Dutch CPGs.**

| Step | Drug 1 | Drug 2 | Drug 3 | Drug 4 | Drug 5 | Drug 6 | Drug 7 | Drug 8 | Number of deviating steps (%) | | |
|---|---|---|---|---|---|---|---|---|---|---|---|
| | | | | | | | | | First-line drugs | Second-line drugs | Across the CPG |
| United Kingdom | | | | | | | | | | | |
| BTS/SIGN | O2 | Nebulized salbutamol | Nebulized ipratropium | Oral CS | Nebulized MgSO4 | Intravenous MgSO4 | Intravenous salbutamol | Intravenous aminophylline | First-line drugs | Second-line drugs | Across the CPG |
| UK1 | 0 | 0 | 0 | 1 | 1 | 0* | 0* | 0* | 2(25%) | 0* | 2(25%) |
| UK2 | 0 | 0 | 0 | 1 | 1 | 0* | 0* | 0* | 2(25%) | 0* | 2(25%) |
| UK3 | 0 | 0 | 0 | 0 | 1 | 0* | 0* | 0* | 1(13%) | 0* | 1(13%) |
| UK4 | 0 | 1 | 0 | 1 | 1 | 0* | 0* | 0* | 0 | 0* | 3(38%) |
| UK5 | 0 | 0 | 0 | 1 | 1 | 0* | 0* | 0* | 2(25%) | 0* | 2(25%) |
| UK6 | 0 | 0 | 1 | 1 | 1 | 0* | 0* | 0* | 3(38%) | 0* | 3(38%) |
| UK7 | 0 | 0 | 0 | 0 | 0 | 0* | 0* | 0* | 0 | 0* | 0 |
| No. of deviating steps across local British CPGs (%) | 0 | 1(14%) | 1(14%) | 5(71%) | 6(86%) | 0 | 0 | 0 | **13(37%)** | 0* | **13(23%)** |
| Netherlands | | | | | | | | | | | |
| NVK | O2 | Nebulized salbutamol | Nebulized ipratropium | Oral CS | None | Intravenous MgSO4 | Intravenous salbutamol | No drug recommended | First-line drugs | Second-line drugs | Across the CPG |
| NL1a | 0 | 0 | 1 | 0 | - | 0 | 0 | 1 | 1(25%) | 1(33%) | 2(29%) |
| NL1b | 0 | 0 | 0 | 0 | - | 0 | 0 | 0 | 0 | 0 | 0 |
| NL2 | 0 | 0 | 0 | 1 | - | 0 | 0 | 1 | 1(25%) | 1(33%) | 2(29%) |
| NL3 | 0 | 0 | 0 | 0 | - | 0 | 0 | 0 | 0 | 0 | 0 |
| NL4 | 0 | 0 | 0 | 0 | - | 0 | 0 | 0 | 0 | 0 | 0 |
| NL5 | 0 | 0 | 0 | 0 | - | 0 | 0 | 0 | 0 | 0 | 0 |
| NL6 | 0 | 0 | 0 | 0 | - | 0 | 0 | 0 | 0 | 0 | 0 |
| NL7 | 0 | 0 | 0 | 0 | - | 0 | 0 | 0 | 0 | 0 | 0 |
| No. of deviating steps across local Dutch CPGs (%) | 0 | 0 | 1(14%) | 1(14%) | | 0 | 0 | **2(29%)** | **2(6%)** | **2(11%)** | **4(8%)** |

*: Local British guidelines received zero points at that position given the absence of a rigid sequence outlined by BTS/SIGN. BTS/SIGN: British Society/Scottish Intercollegiate Guidelines Network; CPG: clinical practice guideline; CS: corticosteroids; MgSO4: magnesium sulfate. NL: Netherlands; O2: oxygen UK: United Kingdom.

two Dutch and six British local CPGs recommended a bolus of intravenous salbutamol before starting a continuous salbutamol infusion.

Apart from recommending nebulized terbutaline (UK4), and omitting nebulized magnesium sulfate (all CPGs but UK7), local British CPGs advised the same sequence of first-line drugs as BTS/SIGN. While they did not strictly differ from national guidance given the lack of a defined second-line drug sequence, local British CPGs, by contrast, recommended four distinct sequences of second-line treatments encompassing the same three drugs. The most common sequence was intravenous magnesium sulfate followed by intravenous salbutamol followed by intravenous aminophylline (UK2, UK4, UK5). By contrast, five out of seven local Dutch CPGs recommended the same treatment sequence as the NVK for both first-line and second-line drugs.

Overall, local British CPGs deviated from the drug choice and order recommended in national guidance (where an order was declared) for 23% of their recommended interventions compared to 8% across local Dutch CPGs (see Table 2). Deviations from national guidance

were more frequent for second-line treatments than for first-line drugs in Dutch local CPGs (11% compared to 6% deviating steps). This was not the case in British local CPGs given the absence of a rigidly defined second-line drug sequence in BTS/SIGN guidance. Five local British CPGs and two Dutch local CPGs differed from their respective national CPG for multiple treatment steps. Local CPGs from both countries featured more variation in second-line interventions than first-line treatments.

Both Dutch and British local CPGs adopted a more flexible tone of recommendation for second-line interventions than for first-line drugs. This was especially true for British local CPGs. Local CPGs from both countries outlined more rigid drug sequences than national CPGs, with more drug recommendations communicated using an imperative tone ('give') and fewer alternatives provided.

## Variation in intravenous salbutamol dosage

Recommended dose ranges for continuous salbutamol infusions across the sampled CPGs are shown in Fig 2. Both the recommended starting doses and maximum doses varied considerably across guidelines. Dutch CPGs advised significantly lower starting doses (in most cases 0.1 μg/kg/min) than British CPGs (1–2 μg/kg/min) and allowed for higher maximum doses (10 μg/kg/min), compared to British CPGs (5–8 μg/kg/min). The salbutamol dosages

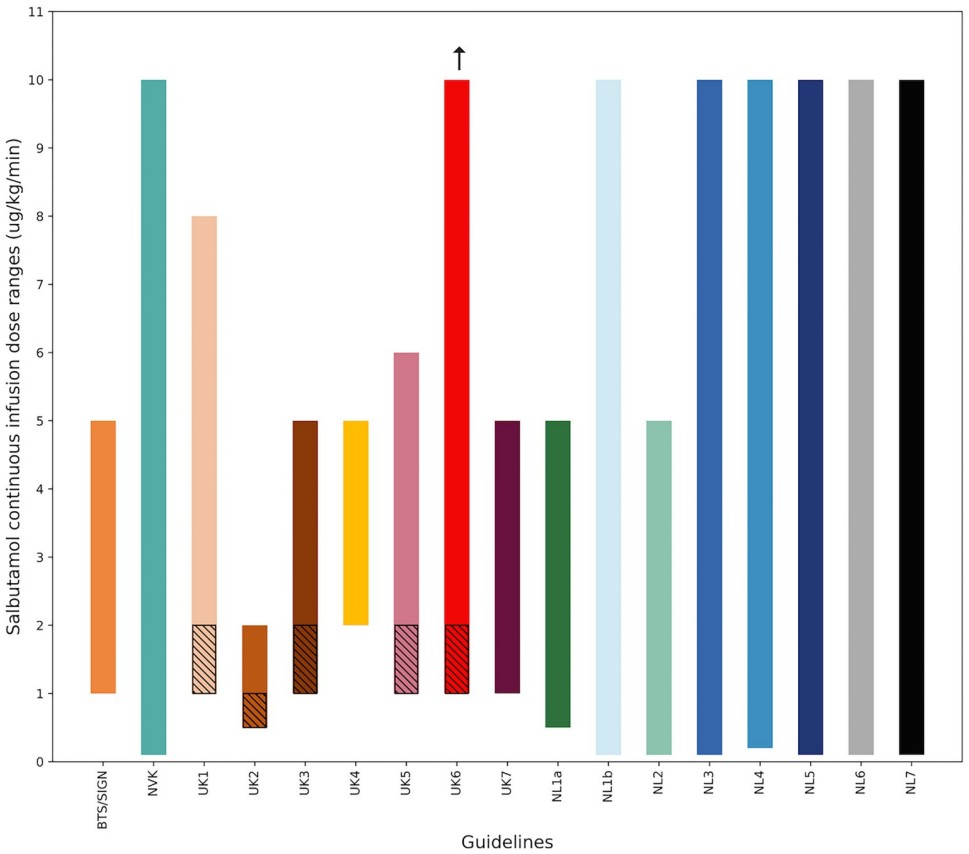

**Fig 2. Recommended dose ranges for continuous salbutamol infusions across sampled CPGs from the UK and the Netherlands.** Hatching indicates starting dose ranges. The arrow indicates the absence of a maximum dose recommended by the concerned guideline. UK2 and UK5: maximum doses calculated for a child of 10 kilograms based on the absolute maximum dosages recommended by each CPG.

**Table 3. Scaled AGREE II domain scores of sampled national and local APA guidelines from the Netherlands and the UK (%).**

| Guideline | 1. | 2. | 3. | 4. | 5. | 6. |
| --- | --- | --- | --- | --- | --- | --- |
| | Scope & purpose | Stakeholder involvement | Rigour of development | Clarity of presentation | Applicability | Editorial independence |
| NVK | 47 | 42 | 78 | 44 | 10 | 13 |
| BTS/SIGN | 78 | 83 | 92 | 72 | 71 | 83 |
| NL1a | 19 | 0 | 20 | 42 | 2 | 0 |
| NL1b | 50 | 6 | 9 | 44 | 13 | 0 |
| NL2 | 69 | 19 | 2 | 61 | 15 | 0 |
| NL3 | 50 | 36 | 7 | 50 | 10 | 0 |
| NL4 | 36 | 0 | 5 | 36 | 4 | 0 |
| NL5 | 47 | 42 | 78 | 44 | 10 | 13 |
| NL6 | 31 | 0 | 0 | 33 | 8 | 0 |
| NL7 | 42 | 6 | 6 | 47 | 10 | 0 |
| **Mean score local Dutch CPGs** | 43 | 14 | 16 | 45 | 14 | 2 |
| UK1 | 36 | 8 | 2 | 28 | 15 | 0 |
| UK2 | 39 | 11 | 7 | 31 | 13 | 0 |
| UK3 | 53 | 22 | 15 | 42 | 8 | 0 |
| UK4 | 56 | 22 | 16 | 56 | 13 | 0 |
| UK5 | 53 | 14 | 5 | 42 | 29 | 0 |
| UK6 | 44 | 17 | 4 | 36 | 15 | 0 |
| UK7 | 53 | 42 | 14 | 50 | 27 | 0 |
| **Mean score local British CPGs** | 48 | 19 | 9 | 40 | 25 | 0 |

APA: acute paediatric asthma; BTS/SIGN: British Society/Scottish Intercollegiate Guidelines Network; CPGs NL: Netherlands; NVK: Nederlandse Vereniging voor Kindergeneeskunde (Dutch Society for Paediatrics); UK: United Kingdom. NL5 corresponds to the NVK guideline.

recommended in two of seven British local CPGs and six of seven Dutch local CPGs aligned with their respective national CPGs.

**Guideline quality.** As described in Table 3, national guidelines from the UK and the Netherlands scored highest on rigour of development (92% and 78% respectively for BTS/SIGN and NVK guidance) and lowest on applicability (71% and 10% respectively for BTS/SIGN and NVK guidance). Local British guidelines and Dutch guidelines scored highest on scope & purpose (mean score of 48% across local British CPGs and 43% across local Dutch CPGs) and clarity of presentation (40% across local British CPGs and 45% across local Dutch CPGs). Local CPGs from both countries scored lowest on editorial independence (mean scores of 0% and 2% respectively for local British and Dutch CPGs). On average, local British guidelines scored lower than BTS/SIGN guidance across all AGREE II domains. Local Dutch guidelines on average scored similarly to the NVK guideline for two domains, namely clarity of presentation and applicability. Average scores for local Dutch and British CPGs were similar across all domains.

## Discussion

This study demonstrated the presence of considerable variation across local CPGs from British and Dutch hospitals for the treatment of children with severe asthma, including numerous deviations from national guidance. Identified discrepancies included the choice, sequence and

recommended methods of administration for the drugs advised for severe APA. Local recommendations for second-line drugs varied more than for first-line interventions, a finding that may reflect the paucity of evidence on effective treatments for severe APA [22]. Differences were also observed in the tone of recommendations made by national CPGs based on this limited evidence. The high level of agreement between national CPGs and their mostly imperative tone on first-line drugs may reflect the ease with which the available evidence could be distilled into a single treatment pathway. Faced with less certain evidence on second-line interventions [33], Dutch national guidance recommended a rigid sequence of interventions while the British national guideline took a more permissive stance, describing various available treatment options without being prescriptive. Possibly as a result of these differences, local British CPGs deviated from national CPGs and from each other, more so than local Dutch CPGs. Substantial variation was also observed in recommended continuous salbutamol infusion rates. The presence of such variation between dosing regimens for one single drug suggests that variation across the sampled CPGs extends beyond the features measured in this analysis.

The AGREE II scores obtained by national guidelines from the UK and the Netherlands were comparable to the ones assigned by Ruszczyński *et al.* to national and professional guidelines for the diagnosis and management of paediatric asthma [34]. The finding of higher scores for scope & purpose and on clarity of presentation and lower scores for editorial independence are in agreement with prior studies of local CPGs [35]. Lower scores may partly result from limited reporting from guideline developers for a given domain [35].

In the presence of multiple sources of clinical guidance [10, 36], local CPGs play a unique role in the translation and dissemination of evidence into local care [14]. In contrast to national CPGs, which are often a careful summary of the best available evidence including all its uncertainties, the design of local CPGs caters to their use as direct guides for action by those clinicians with less experience and time to appraise the evidence themselves as part of their decision-making [37, 38]. As such, local CPGs ought to be immediately intelligible and implementable by these clinicians. This requires simplification of the detailed knowledge contained within national CPGs and the introduction of other sources of knowledge, including expert opinion, if evidence is lacking [1, 39, 40]. Given the slowercycle of national CPG development, and the focus of national guidelines on higher-grade evidence [10, 11], local CPGs may also constitute a preferred vehicle to incorporate new evidence alongside expert insights derived from local practice [16]. The development of local CPGs, furthermore, entails adaptations in the recommendations of national CPGs in order to align with a hospital's resources, patient populations and working processes [12, 14].

Although the intended use of local CPGs may justify deviations from national guidance, some of the variation measured across the sampled local CPGs may not arise from a lack of evidence or the need to adapt national guidance to a given hospital setting [12, 41, 42]. The examined local CPGs, in fact, often failed to declare the methods by which they were produced, or to identify or explain their deviations from national guidance. In the absence of justification such deviations may be considered a source of unwarranted variation in clinical care [42–44].

## Research and policy implications

While local CPGs are widely relied upon as guides to action in a given care setting [14], research into the development of clinical guidelines has focussed on those authored by national and professional bodies. This study highlights the need to further investigate the origins of variation between local CPGs and the effects of CPG variation on variation in clinical care.

Different priorities may characterize national and local CPG development [14], and result in inconsistencies between national and local guidance. Consequently, where local guidelines

deviate from national ones, justification for these deviations should be provided to help clinicians navigate this inconsistency

This study demonstrates that the treatment received by a child with acute asthma may be partly dictated by the hospital they present to. Variation in the treatment of APA, as other conditions, has been linked to poor clinical outcomes [17, 18, 20]. The extent to which this variation represents an appropriate adaptive response to local circumstances remains unknown. One driving factor in the widespread development and use of local guidelines, as revealed by this study, may be the inability of national guidance to achieve sufficient clarity to be used effectively by clinicians [45].

The widespread variation in the recommended treatment of a high-risk condition identified in this study, must be accompanied by open sharing of local CPGs. If excess harm arising from varying treatments for APA (amongst other conditions) is to be reduced, a culture of transparent, inter-organisational learning from local practice should be established. Alongside transparency of recommendations, the outcomes of local centres should be shared to enable the evaluation of the relationship between differences in treatment and differences in clinical outcomes. By sharing both local CPGs and outcomes data between providers, national bodies would be better informed of the use of these critical documents and clinicians would be better equipped to assess the potential benefits of variations to their practice.

## Strengths and limitations

In this study, we applied a novel methodology to analyse and represent variation across a type of clinical guidance that remains understudied. This method could be similarly applied to guidelines for other conditions. This study had several limitations. Because local CPGs tend not to be shared beyond the hospital in which they were developed [14], local guidelines were obtained through convenience sampling. The APA guidelines obtained were from seven out of the eight tertiary paediatric hospitals in the Netherlands and from tertiary hospitals across the UK. As local CPGs were obtained from specialist paediatric centres, they may not be representative of the CPGs used in other less specialised paediatric centres.

While some of the measured variation may be assigned to the varied care settings for which the sampled local CPGs were developed (ranging from emergency departments to critical care units), it is unlikely that the breadth of variation that was measured was attributable to setting alone. In order to represent variation in a legible way across numerous guidelines, differences across local CPGs were described with reference to their respective national guidance. This was done in the absence of a clearly established customary hierarchy between national and local guidelines in the UK or in the Netherlands [45].

Several local British CPGs were last updated prior to the publication of the version of British national guidance that was examined. This may explain some of the deviations that were found. Upon comparison however, the three most recent versions of BTS/SIGN guidance on asthma, published in 2014, 2016 and 2019, did not differ in their recommendations for treating most severe APA [23, 46, 47]. The analysis focused on a single treatment pathway, and did not encompass all grades of APA severity. The drug pathway for most severe APA was chosen as the most extensive drug sequence that could be compared across the sampled guidelines, including all available APA therapies up to maximal escalation. Although this pathway was associated with severity assessment parameters that did not fully align across guidelines, the chosen pathway ensured comparability across the sample for the sickest APA patients. Whereas this study describes substantial variation across local CPGs and between local CPGs and their relevant national guidance, it does not explore the process by which this variation arises or the impact such variation has on clinical outcomes. Lastly, the scores obtained as part

of the AGREE II assessment of guideline quality cannot be interpreted as a proxy for ease of implementation nor adherence to a CPG [36].

## Conclusions

Local British and Dutch CPGs varied substantially in their treatment recommendations for severe acute paediatric asthma and scored poorly on quality. Although limited to a subset of guidelines, these findings may indicate the presence of considerable variation in local CPGs across medical conditions. Variation in the content and quality of local guidelines may contribute to variation of care more broadly and potentially undermine healthcare quality.

## Supporting information

**S1 File.**
(PDF)

## Acknowledgments

The authors would like to acknowledge contribution of Melanie Wilson to data collection and Matthew Harrison to the depiction of data.

## Author Contributions

**Conceptualization:** Charlotte Koldeweij, Nicholas Appelbaum, Carmen Rodriguez Gonzalvez, Joppe Nijman, Jonathan Clarke.

**Data curation:** Charlotte Koldeweij.

**Formal analysis:** Charlotte Koldeweij, Ruud Nijman.

**Investigation:** Charlotte Koldeweij.

**Methodology:** Charlotte Koldeweij, Nicholas Appelbaum, Carmen Rodriguez Gonzalvez, Joppe Nijman, Ruud Nijman, Jonathan Clarke.

**Supervision:** Nicholas Appelbaum, Joppe Nijman, Jonathan Clarke.

**Visualization:** Charlotte Koldeweij.

**Writing – original draft:** Charlotte Koldeweij.

**Writing – review & editing:** Charlotte Koldeweij, Nicholas Appelbaum, Carmen Rodriguez Gonzalvez, Joppe Nijman, Ruchi Sinha, Ian Maconochie, Jonathan Clarke.

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
