## [Decision Letter · Decision Letter 0]

17 Jan 2022

PONE-D-21-30920Mind the gap: Mapping variation between national and local clinical practice guidelines for acute paediatric asthma from the United Kingdom and the NetherlandsPLOS ONE

Dear Dr. Koldeweij,

Thank you for submitting your manuscript to PLOS ONE. After careful consideration, we feel that it has merit but does not fully meet PLOS ONE’s publication criteria as it currently stands. Therefore, we invite you to submit a revised version of the manuscript that addresses the points raised during the review process.

We look forward to receiving your revised manuscript.

Kind regards,

Maria G Grammatikopoulou

Academic Editor

PLOS ONE

Journal Requirements:

"This article is independent research supported by grants from the National Institute for Health Research (NIHR) Imperial Patient Safety and Translational Research Centre (PSTRC) PSTRC_2016_004. Infrastructure support for this work was provided by the NIHR Imperial Biomedical Research Centre (BRC) 1215-20013. JC acknowledges support from EPSRC grant EP/N014529/1 supporting the EPSRC Centre for Mathematics of Precision Healthcare and the Wellcome Trust 215938/Z/19/Z. Funding organisations were not involved in the design of the study and collection, analysis, and interpretation of data or in writing the manuscript."

"Two authors (NA and CK) had support from the National Institute for Health Research (NIHR, accessible from https://www.nihr.ac.uk/) Imperial Patient Safety and Translational Research Centre (PSTRC) PSTRC_2016_004. Infrastructure support for this work was provided by the NIHR Imperial Biomedical Research Centre (BRC) 1215-20013. JC acknowledges support from the Engineering and Physical Sciences Research Council (EPSRC, accessible from: https://epsrc.ukri.org/) grant EP/N014529/1 supporting the EPSRC Centre for Mathematics of Precision Healthcare and from the Wellcome Trust (accessible from: https://wellcome.org/), grant 215938/Z/19/Z. The funders had no role in study design, data collection and analysis, decision to publish, or preparation of the

manuscript."

Additional Editor Comments (if provided):

Dear authors,

thank you for your submission. Please find attached the reviewer's comments, we are looking forward to your revision!

Reviewers' comments:

Reviewer's Responses to Questions

**Comments to the Author**

1. Is the manuscript technically sound, and do the data support the conclusions?

Reviewer #1: Yes

Reviewer #2: Partly

2. Has the statistical analysis been performed appropriately and rigorously? 

Reviewer #1: Yes

Reviewer #2: N/A

3. Have the authors made all data underlying the findings in their manuscript fully available?

Reviewer #1: Yes

Reviewer #2: No

4. Is the manuscript presented in an intelligible fashion and written in standard English?

Reviewer #1: Yes

Reviewer #2: Yes

5. Review Comments to the Author

Reviewer #1: The authors performed a study evaluating the variation between national and local clinical practice guidelines for acute paediatric asthma in two countries. The study is well written and the topic is interesting. However, I have some methodological concerns regarding the sampling method.

The authors used a convenience sampling method to obtain local and national guidelines. How do they handle the importing biases of this technique?

Please include all the abbreviations in footnote of Table 1.

It would be interesting also to create a new column in Table 1 with the AGREE II overall score of each guideline. Apart from variations in treatment recommendations, the quality and reporting of practice guidelines is of great importance, as it will aid pediatric asthma practitioners to select the highest quality guidelines.

The following review on quality appraisal of preschool wheezing and asthma guidelines in children should be cited and discussed: https://pubmed.ncbi.nlm.nih.gov/32816386/

Authors should include the convenience sampling method as a limitation of their study.

Reviewer #2: The study's central idea is interesting, but the sample is limited and chosen by convenience, which weakens the study. It is a local study, of little interest on a global scale, and may be submitted to a journal of regional interest. The text can be improved, making it more objective and fluid, making it easier to read. The discussion can also be deepened, carrying out a critical analysis and proposals for changes.

6. PLOS authors have the option to publish the peer review history of their article (what does this mean?). If published, this will include your full peer review and any attached files.

Reviewer #1: No

Reviewer #2: **Yes: **Marina de Barros Rodrigues

---

## [Author Response · Author response to Decision Letter 0]

21 Mar 2022

Dear Editor and Reviewers, 

Thank you very much for your consideration of our manuscript ‘Mind the gap: Mapping Variation between National and Local Clinical Practice Guidelines for Acute Paediatric Asthma from the United Kingdom and the Netherlands’. We have provided responses to each reviewer comment in turn below. Reviewer 1’s important suggestion we include AGREE II scores for each guideline has led to an increase in word count to describe the methods used, results obtained and to discuss their significance. We have attempted to offset this increase in word count with a tighter edit to other sections of the manuscript. 

Reviewer #1: 

1) The authors performed a study evaluating the variation between national and local clinical practice guidelines for acute paediatric asthma in two countries. The study is well written and the topic is interesting. However, I have some methodological concerns regarding the sampling method. The authors used a convenience sampling method to obtain local and national guidelines. How do they handle the importing biases of this technique? Authors should include the convenience sampling method as a limitation of their study.

Thank you for your helpful review of our study and your kind comments. 

We agree that the method of obtaining local guidelines is subject to selection bias. In the case of our study, the only local guidelines available to the authors were from specialist (tertiary) paediatric centres. It is potentially the case that these local CPGs are not representative of those in use in other less specialized paediatric units. We have acknowledged this as a limitation in the ‘Strengths and limitations’ section of the discussion. 

Local CPGs are not usually publicly available. As researchers we sought to draw upon guidelines that were available to us to begin what we feel is an important discussion about local CPG variability and transparency. We have also raised in the discussion the need for wider transparency in local CPGs to facilitate further study of variation in practice without the unavoidable selection bias of our study. 

The national guidelines used in the study are the two widely used guidelines produced by the respective learned bodies of the UK and Netherlands and were therefore chosen as a single reference point for national recommendations. 

2) It would be interesting also to create a new column in Table 1 with the AGREE II overall score of each guideline. Apart from variations in treatment recommendations, the quality and reporting of practice guidelines is of great importance, as it will aid pediatric asthma practitioners to select the highest quality guidelines. The following review on quality appraisal of preschool wheezing and asthma guidelines in children should be cited and discussed: https://pubmed.ncbi.nlm.nih.gov/32816386/

Thank you for this very helpful comment. In light of your suggestion, we have conducted a review of each national and local CPG according to the AGREE II instrument and have provided scores and interpretation for each domain of the AGREE II, now summarized in Table 3 of the revised manuscript. We have additionally discussed and cited the very helpful suggested review. Rather than use overall AGREE II scores for the study, we have instead reported on individual domains of AGREE II as we did not want to mask variation in scores within domains that may be insightful for the reader. 

3) Please include all the abbreviations in footnote of Table 1.

Thank you for this. The abbreviations were added in a footnote of Table 1. 

Reviewer #2: 

1) The study's central idea is interesting, but the sample is limited and chosen by convenience, which weakens the study. 

Thank you for this comment. We agree that selection bias in the inclusion of local guidelines is a limitation of this study which may impact upon the validity and generalizability of findings. We have made changes to the discussion of the manuscript to reflect this. This is also expanded upon in response to the first comment of Reviewer 1. 

2) It is a local study, of little interest on a global scale, and may be submitted to a journal of regional interest. 

Thank you for this comment. We acknowledge that this study is somewhat limited in its focus upon two countries with largely well-resourced health systems. However, in our decision to submit to PLoS One, we were conscious of its policy to evaluate research independently of its perceived regional or global significance. 

Additionally, previous work by our group (doi: 10.1186/s12916-021-01963-0) has identified variation in national clinical practice guidelines to be a global phenomenon, yet work examining the variation in local clinical practice guidelines is very limited considering the scale of their use in everyday clinical practice. This study provides a potential framework by which other researchers may examine local CPGs in their country or clinical specialty. 

3) The text can be improved, making it more objective and fluid, making it easier to read. 

Thank you for this comment. All sections of the manuscript have been revised to improve the fluency of the text. 

4) The discussion can also be deepened, carrying out a critical analysis and proposals for changes.

The discussion was revised in light of the findings from the AGREE II guideline quality assessment suggested by Reviewer 1, including suggestions for improving current practices around national and local clinical practice guideline development.

Best wishes, 

Charlotte Koldeweij, on behalf of all the authors

---

## [Decision Letter · Decision Letter 1]

11 Apr 2022

Mind the gap: Mapping variation between national and local clinical practice guidelines for acute paediatric asthma from the United Kingdom and the Netherlands

PONE-D-21-30920R1

Dear Dr. Koldeweij,

We’re pleased to inform you that your manuscript has been judged scientifically suitable for publication and will be formally accepted for publication once it meets all outstanding technical requirements.

Kind regards,

Maria G Grammatikopoulou

Academic Editor

PLOS ONE

Additional Editor Comments (optional):

Reviewers' comments:

Reviewer's Responses to Questions

**Comments to the Author**

1. If the authors have adequately addressed your comments raised in a previous round of review and you feel that this manuscript is now acceptable for publication, you may indicate that here to bypass the “Comments to the Author” section, enter your conflict of interest statement in the “Confidential to Editor” section, and submit your "Accept" recommendation.

Reviewer #1: All comments have been addressed

Reviewer #2: All comments have been addressed

2. Is the manuscript technically sound, and do the data support the conclusions?

Reviewer #1: Yes

Reviewer #2: Yes

3. Has the statistical analysis been performed appropriately and rigorously? 

Reviewer #1: Yes

Reviewer #2: Yes

4. Have the authors made all data underlying the findings in their manuscript fully available?

Reviewer #1: Yes

Reviewer #2: Yes

5. Is the manuscript presented in an intelligible fashion and written in standard English?

Reviewer #1: Yes

Reviewer #2: Yes

6. Review Comments to the Author

Reviewer #1: (No Response)

Reviewer #2: (No Response)

7. PLOS authors have the option to publish the peer review history of their article (what does this mean?). If published, this will include your full peer review and any attached files.

Reviewer #1: No

Reviewer #2: **Yes: **Marina de Barros Rodrigues

---

## [Editor Report · Acceptance letter]

9 May 2022

PONE-D-21-30920R1 

Mind the gap: Mapping Variation between National and Local Clinical Practice Guidelines for Acute Paediatric Asthma from the United Kingdom and the Netherlands 

Dear Dr. Koldeweij:

I'm pleased to inform you that your manuscript has been deemed suitable for publication in PLOS ONE. Congratulations! Your manuscript is now with our production department. 

Kind regards, 

on behalf of

Dr. Maria G Grammatikopoulou 

Academic Editor

PLOS ONE